# `miniCTX`: Neural Theorem Proving with (Long-)Contexts

**Jiewen Hu    Thomas Zhu    Sean Welleck**
Carnegie Mellon University

## Abstract

Real-world formal theorem proving often depends on a wealth of context, including definitions, lemmas, comments, file structure, and other information. We introduce `miniCTX`, which tests a model's ability to prove formal mathematical theorems that depend on new context not encountered in training. `miniCTX` contains theorems sourced from real Lean projects and textbooks, each associated with a context that can span tens of thousands of tokens. Models are tasked with proving a theorem given access to code from the theorem's repository, which contains context that is needed for the proof. As a baseline for `miniCTX`, we tested fine-tuning and prompting methods that condition theorem proving on preceding context. Both approaches substantially outperform traditional methods that rely solely on state information. We found that this ability to use context is not captured by previous benchmarks such as `miniF2F`. Alongside `miniCTX`, we offer NTP-TOOLKIT for automatically extracting and annotating theorem proving data, making it easy to add new projects into `miniCTX` to ensure that contexts are not seen during training. `miniCTX` offers a challenging and realistic evaluation of neural theorem provers.

## 1   Introduction

Formal theorem proving in interactive theorem provers (ITPs) provides a testbed for evaluating the reasoning capabilities of large language models (LLMs). Theorem proving capabilities can then directly translate to automation for mathematicians, such as via tools that complete or formalize proofs [1–4]. However, despite their promise, we see a gap between the evaluation of current language model-based provers and the complexity of real-world theorem proving.

Our motivating observation is that theorems and proofs depend on various forms of *context*, such as newly-defined definitions and lemmas. For instance, to prove results about a square, one might first formalize a definition of a rectangle, prove some results about rectangles, then specialize them to a newly-defined square [5] (Figure 1). However, existing methods for training and evaluating LLM-based theorem provers often fail to incorporate the full range of contextual information available in real-world projects. For example, benchmarks often focus on proving standalone competition problems (e.g., `miniF2F` [6]) or theorems from a library that the model has trained on (e.g., Mathlib [7, 8]), and state-of-the-art LLM-based provers are trained to accept only a proof state as input, making them unaware of new theorems and definitions [9–11]. While some existing work, including premise selection techniques [12, 13, 8] and datasets like CoqGym [14], have explored theorem proving based on information beyond the current state, they often focus on a subset of the available information. They primarily focus on providing relevant premises, such as lemmas, to assist in proof construction.

Building on these foundations, we propose `miniCTX`: a benchmark that seeks to expand the scope of context used in theorem proving. We extend beyond traditional premise selection explored in prior benchmarks (e.g., [8, 14]) by incorporating a more comprehensive set of contextual elements. This includes premises, but also prior proofs, comments, notation, and structural components like imports and declarations. By doing so, `miniCTX` aims to drive the development of methods that understand

Table 1: Comparison of theorem proving benchmarks across several key features.

| Benchmark | Language | Premise | Full Context | Multi-source | Temporal Split |
|-----------|----------|---------|--------------|--------------|----------------|
| miniF2F [6] | Multiple | ✗ | ✗ | ✗ | ✗ |
| ProofNet [15] | Lean | ✗ | ✓ | ✓ | ✗ |
| LeanDojo [8] | Lean | ✓ | ✗ | ✗ | ✗ |
| LeanStep [7] | Lean | ✓ | ✗ | ✓ | ✗ |
| CoqGym [14] | Coq | ✓ | ✗ | ✓ | ✗ |
| PISA [16] | Isabelle | ✗ | ✗ | ✓ | ✗ |
| miniCTX (Ours) | Lean | ✓ | ✓ | ✓ | ✓ |

and work with context that occurs in complex, real-world theorem proving tasks. Additionally, considering the common use of pre-trained language models we mitigate potential data contamination by continually and automatically updating `miniCTX` with new Lean projects, so that evaluated theorems are not seen during training. Our key contributions are:

`miniCTX` **Benchmark:** We introduce `miniCTX`, the first benchmark designed specifically to evaluate theorem proving in real-world settings where proofs depend on in-file definitions, lemmas, and context from formal projects. `miniCTX` presents a unique challenge by requiring models to reason over long contexts and handle dependencies that arise in real-world theorem proving tasks.

**NTP-TOOLKIT:** To facilitate the automatic updating of `miniCTX`, we developed the NTP-TOOLKIT, which automatically extracts relevant theorems and contexts from Lean projects. Additionally, we provide a Lean REPL wrapper that enables simpler evaluation on `miniCTX`.

**Baseline Evaluations:** We evaluate `miniCTX` on several existing baseline models, including different fine-tuning and prompting strategies, as well as premise selection. We also propose file-tuning, a strong baseline method for training models using full file contexts, where both the theorem statements and their surrounding context are provided during training. This approach establishes a robust baseline for future work on context-dependent theorem proving.

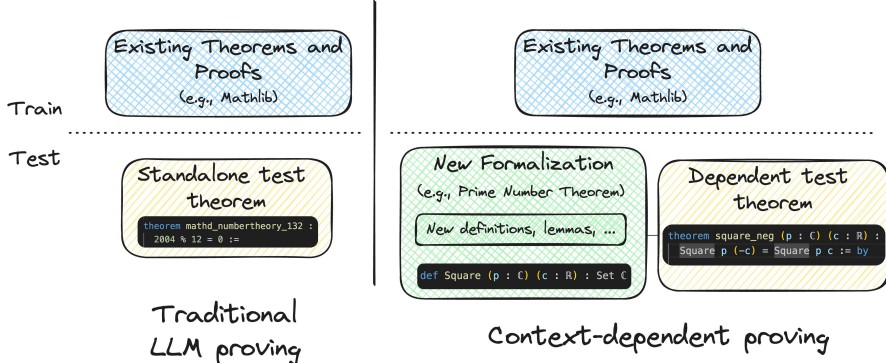

Figure 1: Many state of the art provers are trained on a static dataset of theorems and proofs, then evaluated on standalone problems such as competition problems (left). We argue that neural provers must also operate in the realistic *context-dependent* setting, in which results depend on working with new mathematical objects and their facts, notations, and the structural elements of the project (imports, variables, etc.) (right).

## 2 Theorem proving with context

Formal theorem proving involves two stages: defining mathematical objects and facts relevant to the desired result, then stating and proving the result itself. Many current language model-based provers focus on the proving process and are trained on static datasets that only use a fixed set of definitions. As a result, they lack the ability to recognize new definitions or lemmas at test time (Figure 1).

**Context-dependent proving.** We study *context-dependent theorem proving*, where the goal is for a model to generate proofs $y$ for new theorems $x$, based on a context $c$ that includes background information such as definitions, lemmas, or natural language comments. Formally, the problem is

$$\text{maximize}_M \ \mathbb{E}_{(x,c)\sim p}\mathbb{E}_{y\sim M(\cdot|x,c)}v(x,c,y), \tag{1}$$

where $(x,c) \sim p$ denotes a (theorem, context) pair from a context distribution $p$, $M$ is a model that produces a proof $y$, and $v$ returns 1 if the proof is correct and 0 otherwise.

We choose Lean [17] as the verifier $v$, because of the large body of recent theorems in Lean that can be used as evaluation data, and the abundance of proving methods in Lean that we use as baselines. We treat a Lean repository as the distribution $p$. Each context $c$ is a subset of the repository, including new definitions, lemmas, notations, imports, and comments that are relevant to the theorem.

## 3 `miniCTX`: a benchmark for theorem proving with context

We develop `miniCTX`, a Lean 4 theorem proving benchmark of theorems that depend on newly-defined lemmas, definitions, and proofs from within a project. `miniCTX` is currently based on 376 theorems from four projects: (1) Prime Number Theorem (**Prime**) [18], (2) Polynomial Freman-Ruzsa Conjecture (**PFR**) [19], (3) an introductory text on theorem proving (**HTPI**) [20], (4) recent results from the standard mathematical library (**Mathlib**) [21] (motivation and details in §D), (5) high energy physics formalization in HepLean (**HEP**) [22], and (6) scientific computing formalizations (**SciLean**) [23].

Each theorem in `miniCTX` consists of the theorem statement, preceding file contents up to the theorem statement, and metadata, in JSON (see §E.1).

1. Theorem statement,
2. Preceding file contents up to the theorem statement,
3. Metadata, including file name, commit and time which the theorem was added, position and length of the theorem and proof, and the number and types of premises used in the human-written proof.

Using our benchmark, users can easily reconstruct the complete context for each theorem, including both in-file and cross-file context. The in-file context is provided directly by preceding file contents, while the cross-file context can be reconstructed using the metadata, which includes information on imported modules. We open-source the dataset and evaluation code.

### 3.1 Key features and challenges

`miniCTX` introduces several key features that distinguish it from other theorem proving benchmarks, addressing challenges that have not been tackled by previous benchmarks:

**Real-world theorem proving.** Unlike popular benchmarks (e.g., miniF2F [6], ProofNet [15], FIMO [24]) that focus on isolated competition problems, real-world research-level theorem proving is heavily dependent on rich mathematical contexts. Therefore, `miniCTX` includes real-world, complex theorems from a variety of ongoing Lean projects, such as Prime Number Theorem (Prime) and Polynomial Freiman-Ruzsa Conjecture (PFR). They rigorously test a model's ability in real-world formalization projects. This diversity contrasts with the LeanDojo benchmark [8], which focuses solely on Mathlib, enabling `miniCTX` to better test a model's generalization in different settings.

**Contextual evaluation.** Proving a theorem often depends on new definitions, lemmas, or other contextual information, which a model may not have seen during training. `miniCTX` includes theorems along with this new context. During evaluation, the model is expected to leverage the provided new context to help prove the theorem.

Beyond previous datasets like LeanDojo [8] and CoqGym [14], which include relevant definitions and theorems, `miniCTX` includes additional useful contextual information that may make some theorems *easier* to prove compared to standalone theorems. For instance, Lean source code can have natural language comments that may help constrain the space of possible proofs. Moreover, some proofs within a file often have analogous patterns or structure, which may make subsequent theorems easier to prove (see §E.2). These additional forms of context occur in the real-world process of formalization, yet their use in neural theorem proving is underexplored.

**Automatically updating the benchmark.** Most modern neural theorem provers use a large language model as a backbone. Therefore, it is crucial to ensure that evaluation content is not seen during (pre-)training, a problem not addressed by previous benchmarks. miniCTX's format is amenable to periodically updating the benchmark with new projects to ensure that proofs are not seen by language models trained prior to a particular date. Future periodic updates will be automatically extracted from new Lean projects and commits using NTP-TOOLKIT (§**??**). See Figure 2 for an illustration.

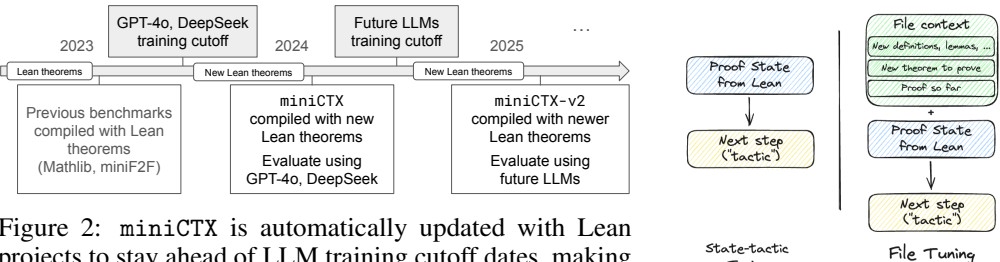

Figure 2: miniCTX is automatically updated with Lean projects to stay ahead of LLM training cutoff dates, making it a suitable benchmark for real-world theorem proving for pre-trained models.

Figure 3: State-tactic vs. file tuning.

## 4   Experiments

### 4.1   Baselines

We evaluate several baselines on miniCTX, demonstrating the importance of context in real-world theorem proving. Our investigation reveals several open challenges that we discuss in §A. See §B for a detailed description of the motivation, baselines, data extraction, and evaluation setup, and §H for full results and more detailed analysis. The baselines are as follows:

**Prompting LLMs.** We first test the ability of a state of the art API-based model, GPT-4o, to generate the complete proof in one pass (pass@1) given the theorem statement, with several few-shot examples provided for guidance. We additionally test whether adding context in the form of preceding file contents improves the proof rate of GPT-4o.

**State-tactic prompting.** Another common approach to theorem proving using language models is to let the model generate a tactic given the current proof state [7–9, 25]. Therefore, we test the *state-tactic prompting* setting, which prompts a model specialized for mathematical tasks, Llemma-7b [26], to output a tactic given a proof state. At test time, the model generates one tactic at a time, and we use a best-first search to construct full proofs [7–9, 1].

**State-tactic tuning.** We follow this *state-tactic* framework and fine-tune a *state-tactic tuned* model from DeepSeek-Coder-1.3b [27] to input proof states and output tactics, trained on human-written tactics in Mathlib, the main mathematical library in Lean, extracted by NTP-TOOLKIT.

**File-tuning.** We then test whether supplying context, in the form of preceding file contents, to the model improves performance. Similar to state-tactic tuning, we fine-tune a 1.3b model to generate a tactic based on (proof state, context) pairs, resulting in the *file-tuned* model.

**Premise selection.** To better simulate a complete context and evaluate on project-level generalization, we apply premise selection to extract relevant premises from imported files within the same repository. We use the premise retriever provided by LeanDojo [8] to identify the top 20 most relevant definitions or lemmas from imported modules and append them to the in-file context.

### 4.2   Results

**Context-dependent methods improve theorem proving.** Table 2 shows baseline performances on miniCTX. We see a dramatic improvement for the file-tuned model (trained on full file context) over the state-tactic model (trained only on proof states) (35.94% vs. 19.53%). Similarly, providing the preceding file context, which includes definitions and lemmas, to GPT-4o results in dramatic improvement compared to using just the proof state (27.08% vs. 11.72%). Figure 4 shows the performance of state-tactic tuned model and file-tuned model on problems with in-file dependencies compared to those without. These findings highlight the importance of providing models with rich

Table 2: Performance comparison (%) of different models on `miniF2F` and `miniCTX`.

| Method | miniF2F | miniCTX | | | | | | | |
| | Test | Prime | PFR | $\text{PFR}_{\text{cross}}$ | Mathlib | HTPI | HEP | SciLean | Avg. |
| --- | --- | --- | --- | --- | --- | --- | --- | --- | --- |
| GPT-4o (full proof) | — | 7.06 | 1.85 | 6.98 | 14.00 | 13.33 | 31.15 | 6.52 | 11.72 |
|   + context | — | 31.76 | 5.56 | 34.88 | 26.00 | **17.78** | 49.18 | 17.39 | 27.08 |
|   + context + premise | — | 29.41 | 7.41 | 39.53 | — | 15.56 | 44.26 | 21.74 | 26.82 |
| State-tactic prompting | 28.28 | 20.00 | 5.56 | 0.00 | 16.00 | 0.00 | 31.15 | 19.57 | 14.58 |
| State-tactic tuning | 32.79 | 17.65 | 5.56 | 0.00 | 22.00 | 11.11 | 52.46 | 19.57 | 19.53 |
| File tuning | **33.61** | 40.00 | 5.56 | **44.19** | **34.00** | 15.56 | **60.66** | **45.65** | **35.94** |
|   + premise | — | **42.35** | **11.11** | 16.28 | — | 8.89 | 50.82 | 32.61 | 30.21 |

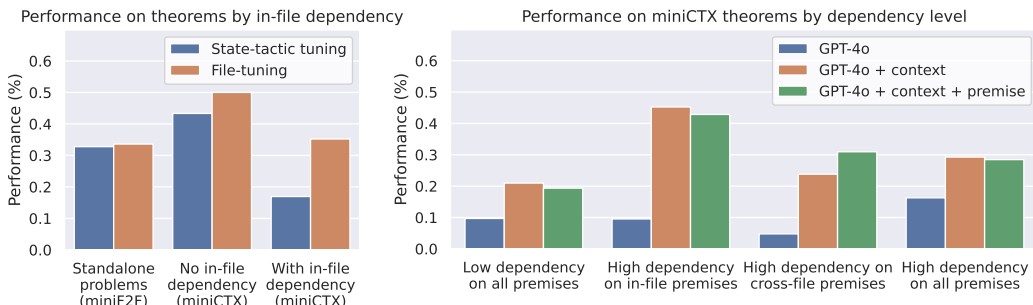

Figure 4: Model performance by dependency on premises. For each theorem in `miniCTX`, we record as metadata whether its human-written proof depends on other definitions or theorems in the same file ("in-file") or in other files ("cross-file"), and test the performance of baselines on each type.

contextual information beyond the immediate proof state, also demonstrating that `miniCTX` is able to measure this ability of context-dependent proving.

**Premise selection improves performance on high cross-file dependency splits.** The results in Table 2 indicate that premise selection has a mixed impact on model performance. For the GPT-4o, premise selection improves performance on high cross-file dependency splits, such as PFR, $\text{PFR}_{\text{cross}}$, and SciLean. This suggests that premise selection helps capture the cross-file context, enabling GPT-4o to make better use of cross-file information. However, for the file-tuned model, premise selection does not consistently improve results, and even performs worse on the $\text{PFR}_{\text{cross}}$ split, which was designed to evaluate the effective use of cross-file premises. Also shown in Figure 4, GPT-4o benefits significantly from premise selection on problems with high cross-file dependencies, but degrades in other cases. This suggests that the retrieved premises differ significantly from the in-file context. Therefore, developing methods that effectively support the integration of cross-file context (e.g., premise selection) alongside in-file context remains an interesting open research direction for improving performance on the `miniCTX` benchmark.

**Evaluation on `miniF2F`.** We evaluate baselines on `miniF2F`, a standard benchmark based on competition problems that do not require context. The file-tuned model improves very little beyond the state-tactic model (33.61% vs. 32.79%), showing that the dramatic difference in context-dependent proving abilities seen on `miniCTX` cannot be captured by `miniF2F`.

**Additional analysis.** Further analysis shows that file-tuning delivers greater gains on problems with stronger dependencies on new lemmas. Both definitions and theorems are crucial in the context, and models show ability to learn proof structure from previous lemmas. See §**??** for more details.

# 5 Conclusion

We studied the realistic setting of proving theorems that depend on new information and project constraints, and formulated an evaluation framework for testing generalization using real Lean projects. We built `miniCTX`, and found that the predominant method for training neural theorem provers fails to enable context dependent proving. Our file tuning method provides a strong starting point for the new challenges opened by our investigation into theorem proving with context.

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

# Appendix

## A  Discussion and future challenges

In addition to general improvements in performance, we comment on some specific open challenges.

**Making better use of long-contexts.** Our file-tuning method simply truncates contexts to be within a token budget (1024), which can discard useful contextual information. We found gains in providing GPT-4o 8,000 tokens of context compared to not providing it context, but its absolute performance was still low. There are several possible strategies that can be explored in future work, including feeding in the entire context, retrieval, or mixtures of the two.

**Repository-level context.** We focused on evaluating in-file context in this paper. As shown in §H.1, many problems require using context outside of the current file. Although we incorporated premise selection as a means of leveraging cross-file context, our experiments indicate that it does not consistently improve performance, even on datasets with high cross-file dependencies. This suggests a need to further investigate how to better integrate premise selection with in-file context. miniCTX provides sufficient metadata to reconstruct the entire environment, allowing for comprehensive investigation into premise selection and other potential methods for leveraging cross-file context.

**Challenging proofs.** Using context through file tuning did not improve performance on the challenging PFR proofs. Moreover, performance is relatively low (19%) on proofs that had a human-written proof of longer than five lines (see §H.2). Proving these kinds of theorems remains an open problem.

**Working with constraints.** As shown in Table 6, model performance drops when the proof cannot use powerful automation tactics. Models have a tendency to invoke these powerful tactics, and struggle with more explicit step-by-step proofs. Improving performance in this setting of miniCTX is an interesting future direction.

## B  Experiment Details

### B.1  Motivation for Baselines

Writing a proof can be seen as a sequential process $(x_1, y_1), (x_2, y_2), \ldots$ of *states* $x_t$ and *tactics* $y_t$. A state contains what is left to prove (the *goal*), and available information (the *hypotheses*). A *tactic* transitions the proof to a new state. If the state contains no remaining goals, the proof is complete. Concretely, a user applies tactics by writing Lean code, Lean keeps track of the state, and the development environment shows the state and the written code.

The traditional approach to training a language model for theorem proving is to train a model on (state, tactic) pairs, i.e., train it to predict the next step of a proof (i.e., the tactic), given the state provided by the proof assistant (i.e., the proof state) [9, 7, 25, 8]. A drawback to this approach is that at test time, the model is not aware of new context outside of the proof state, such as new lemmas. We will see later on that models trained with this state-tactic approach fail at context-dependent proving.

As a stronger baseline for context-dependent proving, we present *file-tuning*, a simple recipe that trains with (preceding file context, proof state, next-tactic) tuples instead of training with (proof state, next-tactic) pairs (Figure 3). This lets the model use new definitions, theorems, or other information that are defined prior to the current tactic at training or at test time. In practice, file-tuning requires extracting contexts and proof states from Lean, which is done by NTP-TOOLKIT.

### B.2  Data extraction

We ran NTP-TOOLKIT's next-tactic extraction on a snapshot of Mathlib, yielding 307,049 examples available at l3lab/ntp-mathlib. We then ran NTP-TOOLKIT's instruction tuning script on these examples, yielding file-tuning examples and state-tactic examples. For the file-tuning examples, as an initial method for handling the long Lean files, we either truncate the middle of an input file so that the file contents is 1024 tokens, or take only the preceding 1024 tokens, with the strategy selected at random for each example. The state-tactic examples are at l3lab/ntp-mathlib-instruct-st. The union of file-tuning and state-tactic examples are at l3lab/ntp-mathlib-instruct-context, split into 583k train, 15k dev, and 15k test examples.

### B.3   Baseline Setups

**File-tuning.**   Next, we fine-tune a language model on the union of file-tuning and state-tactic examples. We use the DeepSeek-Coder-1.3b language model [27] based on its performance on code generation tasks and its size, which allowed us to fine-tune with our computational budget. We fine-tune the model for 3 epochs and select the model based on held-out validation perplexity evaluated every 4,000 steps. The model is available at `l3lab/ntp-mathlib-context-deepseek-coder-1.3b`.

**State-tactic tuning.**   We fine-tune a similar model on `l3lab/ntp-mathlib-instruct-st` using the same hyperparameters. The model is available at `l3lab/ntp-mathlib-st-deepseek-coder-1.3b`.

### B.4   Evaluation setup

We evaluate models for the task of tactic-based theorem proving: given a theorem statement, a model generates one tactic at a time while receiving states from the proof assistant. We use a standard best-first search strategy [9, 7, 28, 8, 1] which prioritizes partial proofs based on the model's average log probabilities. This search method is parameterized by the number of generated tactics per iteration $S$, and the maximum number of iterations $T$. We use the setting from [26, 1] ($S = 32$, and $T = 100$).

We evaluate five types of baselines: (1) pass@1 full proof generation using GPT-4o: we prompt the model with only the theorem statement and require it to generate a complete proof (see Appendix (§G.2) for details of the prompts and few-shot examples); (2) pass@1 full proof generation with in-file context using GPT-4o: we supplement the theorem statement with up to 8000 tokens of in-file context; (3) the **file-tuning model** described in (§4.1); (4) the **state-tactic model** described in (§4.1); and (5) a **state-tactic prompting** model: we prompt a pre-trained language model with (state, tactic) examples, using Llemma-7b [26].

## C   Analysis

We analyze the baseline models on `miniCTX` further along several axes, including the kinds of contextual dependencies, the difficulty, and the content made available in the context.

**File-tuning especially helps on problems with infile dependencies.** We use the `miniCTX` metadata to categorize theorems based on their in-file dependencies. Figure 6 shows the performance of state-tactic tuned model and file-tuned model on problems with in-file dependencies compared to those without. We also show `miniF2F` as an additional reference point for problems without in-file dependencies. The file-tuned model shows a marked improvement over the state-tactic tuned model, especially in problems that have dependencies on context. We conclude that file-tuning specifically helps in the realistic setting of theorem proving with new definitions and theorems in context.

**Premise selection helps but may interfere with in-file context.** We use `miniCTX` metadata to categorize problems based on their cross-file dependencies, evaluating the impact of premise selection across the entire dataset. As shown in Figure 4, GPT-4o benefits significantly from premise selection on problems with high cross-file dependencies, showing improved performance when leveraging relevant premises from imported files. However, we also observe that premise selection can interfere with in-file context, leading to inconsistent results, particularly when the available in-file context is relatively short. This suggests that adding cross-file premises may sometimes disrupt the model's ability to focus on the in-file information. Further analysis of this interference is included in §H.3. This highlights the need for more sophisticated integration strategies that can balance both in-file and cross-file contexts effectively.

**Models can learn from previous proofs in the file context.**   To determine the contribution of different components in the in-file context, we conducted an ablation study on the `PFR.ForMathlib.Entropy.Basic` file, which contains numerous co-related lemmas and rich natural language comments, making it an ideal candidate to investigate the influence of different context components. In this ablation, we systematically removed specific parts of the in-file context and evaluated the model's ability to generate proofs under these modified conditions. As shown in Table 3, both the file-tuned model and GPT-4o benefit from the inclusion of previous proofs in the file context. This indicates that models are capable of learning proof strategies from existing proofs in the file and effectively applying them to new problems (see §H.4 for more examples).

Table 3: Ablation study on different context components for theorem proving.

| Environment | Definitions | Lemma Statement | Lemma Proof | Natural Language Comments | File-tuning | GPT-4o |
|:---:|:---:|:---:|:---:|:---:|:---:|:---:|
| ✗ | ✗ | ✗ | ✗ | ✗ | 14.12% | 8.24% |
| ✓ | ✗ | ✗ | ✗ | ✗ | 25.88% | 2.35% |
| ✓ | ✓ | ✗ | ✗ | ✗ | 24.71% | 9.41% |
| ✓ | ✓ | ✓ | ✗ | ✗ | 27.06% | 22.35% |
| ✓ | ✓ | ✓ | ✓ | ✗ | 32.94% | **34.12%** |
| ✓ | ✓ | ✓ | ✗ | ✓ | 28.24% | 23.53% |
| ✓ | ✓ | ✓ | ✓ | ✓ | **35.29%** | 31.76% |

Table 4: Overview of problem statistics in `miniF2F` and `miniCTX`.

| | Split | Problems | Avg. Context Length (tokens) | Avg. Proof Steps |
|:---|:---|:---:|:---:|:---:|
| `miniF2F` [6] | Valid/Test | 488 | 153* | 3.0[†] |
| | Prime | 87 | 10,630 | 3.6 |
| | PFR | 54 | 17,495 | 27.7 |
| `miniCTX` | Mathlib | 50 | 14,440 | 6.1 |
| | HTPI | 185 | 39,050 | 10.7[†] |
| | **All** | 376 | 26,106 | 10.9 |

*Only counting library imports and definitions.    [†]Excluding theorems without proofs.

**Natural language comments contribute in certain settings.** Our ablation also explored the effect of natural language comments in the in-file context. Though the impact was not dramatic, comments written in natural language were found to be helpful in certain settings. In scenarios where proofs were excluded from the context, adding comments resulted in slight performance gains for both models. For the file-tuned model, these gains were further amplified when proofs were included alongside comments, demonstrating the value of combining formal context with explanatory natural language. However, for GPT-4o, the presence of comments when proofs were included led to a slight decrease in performance, suggesting that effective context selection may vary depending on the model architecture and underlying training characteristics.

**File-tuning improves across all difficulty levels and context lengths.** Finally, Appendix §H.2 shows performance on problems categorized by the length of the human-written proof (when available), which we take as a rough proxy of the problem difficulty. The file-tuned model improved on all three difficulty categories. Appendix §H.2 also shows that file-tuning had improved accuracy across context lengths, particularly for problems with longer contexts. Longer contexts may imply more dependencies, suggesting that these problems can benefit more from file-tuning.

**Models rely on common symbolic automation.** To demonstrate an additional kind of context-dependence, we perform an additional analysis on Math2001 [20], which is another Lean textbook setting.[1] In particular, the textbook code disables powerful automation tactics including `simp` and `linarith` to promote manual reasoning, akin to traditional textbook exercises. For example, Math2001 includes numerous arithmetic problems that are trivial with automation tactics (e.g., `linarith`) but are challenging for models to explicitly prove with step-by-step reasoning (e.g., via `calc`). In Table 6 we evaluate models with the automation disabled, and observe substantial performance drops, confirming the reliance on automation tactics. We also find that the state-tactic tuned model relies on `simp` for unseen definitions, making it performing similarly well to the file-tuned model on theorems that only rely on new definitions (§H.6).

## D  `miniCTX` Source

**Prime Number Theorem.** PrimeNumberTheoremAnd [18] is a project started in January 2024 that formalizes the prime number theorem in Lean as well as related concepts, such as residue calculus on rectangles in $\mathbb{C}$. We find the files `Rectangle.lean` and `ResidueCalcOnRectangles.lean` suitable for our purpose of testing context-dependent theorem proving, especially when we use preceding file content as context, as each file is self-contained within the project and contains new definitions (rectangles, squares) and many interdependent lemmas. See §E.2 for an illustration of such

---

[1]See Appendix F.1 for further details on Math2001. Due to licensing we do not include it in `miniCTX`.

lemmas. In addition, most theorems from `ResidueCalcOnRectangles.lean` rely on the definitions from `Rectangle.lean`, which serves as a perfect example of cross-file dependencies. We extracted 87 theorems from these files. Assuming that a model was trained prior to January 2024, this split guarantees the evaluation of project-level, context-level, and theorem-level generalization.

**PFR.** PFR [19] is a project started in November 2023 that formalizes a proof of the Polynomial Freiman–Ruzsa (PFR) conjecture. We included 54 theorems from PFR. We find that proofs of theorems in PFR tend to be much more monolithic and longer in length than those in Mathlib or other libraries. PFR also defines custom mathematical concepts and notations (such as Ruzsa distance) and a proof typically depends on many lemmas in PFR outside the current file. All of the theorems were added to PFR after November 2023. Assuming that the model was trained prior to this date, this split guarantees the evaluation of project-level, context-level, and theorem-level generalization.

**Recent Mathlib Commits.** Lean's mathematical library, Mathlib [21], is a community-maintained Lean repository including mathematical concepts as well as programming APIs and common tactics. It is the single largest Lean library that users contribute to, and is therefore representative of the production environment in which neural theorem provers are deployed. Mathlib is a long-standing project, and it is common practice to train language model-based provers on Mathlib. It is therefore likely that Mathlib source files have been observed during training. However, Mathlib is frequently updated, with new definitions, theorems, and refactorings occurring on a daily basis. Hence we can test theorem-level generalization by tasking a model with proving *newly added* theorems, given a context that may or may not have been observed during training.

We included 50 theorems added to Mathlib in April 2024, by filtering recent Mathlib commits to ones that only add new theorems. Many of the theorems added are simple lemmas that depend on earlier ones (e.g., ones seen during training). As Mathlib generally refactors new theorems by breaking down long theorems to shorter lemmas, most new theorems are not difficult to prove, and give a realistic representation of where neural theorem provers are used. Assuming that the model was trained prior to April 2024, the Mathlib split guarantees the evaluation of theorem-level generalization.

**HTPI.** HTPI contains the Lean code for the book *How to Prove It* (HTPI) [29], which explains a systematic approach to constructing mathematical proofs with Lean. It covers various topics, including elementary logic, number theory, and proving techniques like mathematical induction, along with their implementation in Lean. As supplementary material to the textbook, the problems in HTPI are formulated in a similar fashion: the files typically start with basic definitions and lemmas that might be used throughout the entire file, followed by exercises and several example problems.[2] Therefore, models can utilize definitions, lemmas, and proof structures from example problems to solve similar exercises. Intuitively, the model must understand and apply the context provided within each file, making it an effective benchmark for testing context-aware theorem-proving models.

## E `miniCTX` Examples

Here we give some examples of the `miniCTX` and its sources to illustrate the format of the data and how and why we collect certain theorems.

### E.1 Example Entry

An entry in the `miniCTX` dataset consists of the theorem statement, preceding file contents, and metadata information. For example, given the following theorem `s_eq_pow_two` in context:

```
import Mathlib.Data.Real.Basic

/-!
# Square function
We define the squaring function `s : ℝ → ℝ` to be `s x := x * x`.
-/
```

---

[2]The main chapter and exercises are separated in the original project: HTPILeanPackage. We manually merged them for evaluation in the fork: HTPILeanPackage4.7.

```
def s (x : ℝ) : ℝ := x * x

lemma s_eq_pow_two {x : ℝ} : s x = x ^ 2 := by
  rw [s, pow_two]
```

We collect its data into `miniCTX`, formatted in JSON as follows:

```
{
  # Preceding file content
  "srcContext": "import␣Mathlib.Data.Real.Basic\\n\\n/-!\\n#␣Square␣function\\nWe␣
      define␣the␣squaring␣function␣`s␣:␣\\u211d␣\\u2192␣\\u211d`␣to␣be␣`s␣x␣:=␣x␣*␣
      x`.\\n-/\\n\\ndef␣s␣(x␣:␣\\u211d)␣:␣\\u211d␣:=␣x␣*␣x\\n\\n",

  # Theorem statement
  "theoremStatement": "lemma␣s_eq_pow_two␣{x␣:␣\\u211d}␣:␣s␣x␣=␣x␣^␣2",

  # Fully qualified theorem name
  "theoremName": "s_eq_pow_two",

  # Temporal metadata
  "fileCreated": "(git␣commit)",
  "theoremCreated": "(git␣commit)",

  # Source metadata
  "file": "MyProject/Square.lean",
  "module": "MyProject.Square",
  "positionMetadata": {
    # Line number the theorem is on
    "lineInFile": 10,
    # Number of tokens before the theorem
    "tokenPositionInFile": 152,
    # Number of premises (definitions, theorems) before the theorem
    "theoremPositionInFile": 1
  },

  # Dependency metadata
  "dependencyMetadata": {
    # Number of definitions or lemmas defined in this file that the theorem uses
    "inFilePremises": true,
    "numInFilePremises": 1,
    # Number of definitions or lemmas defined in this repository that the theorem
        uses (including in-file ones)
    "repositoryPremises": true
    "numRepositoryPremises": 1,
    # Number of total premises (in file, repository, or otherwise)
    "numPremises": 2,
    # Modules imported in the current file
    "importedModules": ["Mathlib.Data.Real.Basic", ...]
  },

  # Proof metadata
  "proofMetadata": {
    "hasProof": true,
    "proof": "by\\n␣␣rw␣[s,␣pow_two]",
    "proofType": "tactic",
    "proofLengthLines": 2,
    "proofLengthTokens": 20
  }
}
```

In additional to individual entries, we also record the version (git commit) of the repository.

### E.2 Prime Number Theorem example

We collect theorems from the `Rectangle.lean` file in PrimeNumberTheoremAnd. The following excerpt from `Rectangle.lean` demonstrates the scenario that often arises in a theorem proving environment where context is critical to producing a proof:

```
import Mathlib.Analysis.Complex.CauchyIntegral
import Mathlib.Analysis.Complex.Convex

open Complex Set Topology

open scoped Interval

variable {z w : ℂ} {c : ℝ}

/-%%
\begin{definition}\label{Rectangle}\lean{Rectangle}\leanok
A Rectangle has corners $z$ and $w \in \C$.
\end{definition}
%%-/
/-- A `Rectangle` has corners `z` and `w`. -/
def Rectangle (z w : ℂ) : Set ℂ := [[z.re, w.re]] ×ℂ [[z.im, w.im]]

namespace Rectangle

lemma symm : Rectangle z w = Rectangle w z := by
  simp [Rectangle, uIcc_comm]

lemma symm_re : Rectangle (w.re + z.im * I) (z.re + w.im * I) = Rectangle z w := by
  simp [Rectangle, uIcc_comm]
```

When proving the final lemma `symm_re`, a model can benefit much from the preceding file contents, which include (1) the existing imports from Mathlib, `variable` declarations, and open namespaces that provide a syntactic context for this theorem, (2) the new definition `Rectangle` in the context, which the model has not seen in training, (3) natural language and LaTeX documentation of the file and `Rectangle` definition, (4) the analogous (in this case identical) proof of the preceding theorem `symm`. We demonstrate that performance on `Rectangle.lean` is indeed much higher when preceding file contents are given as context to a model.

For future data added to `miniCTX` that specifically test the preceding file contents as context, we will ensure it is standalone like `Rectangle.lean`, i.e. it does not import any other unseen files from the same repository, so the preceding file contents already contain all important information relevant to the proof.

## F  Additional datasets

In addition to problems in `miniCTX`, we also evaluated other datasets that are not included due to copyright reasons.

### F.1  Math2001

Math2001 [20] contains the Lean code for the book *The Mechanics of Proof* by Heather Macbeth, an introductory text on mathematical theorem proving with accompanying Lean code. Each chapter of The Mechanics of Proof covers an introductory topic and walks through how to write the associated mathematics in Lean, along with exercises. The topics include proofs by calculation, proofs with structure, parity and divisibility, logic, induction, number theory, functions, sets, and relations. A unique aspect of Math2001 is that it disables common Lean automation for pedagogical purposes. For example, a student must write out an equality proof in detail, with each step justified. It also defines new tactics and definitions separate from the common Lean libraries. Typically a file in the textbook will show examples of such proofs, followed by exercises for a student to complete. We can view this as a form of contextual adaptation: a model must prove the theorem according to the constraints of the textbook. Math2001 has 41 files that include examples and exercises. We selected 1

| Models | Math2001 |
|---|---|
| GPT-4o (full proof) | 11.76% |
| GPT-4o (+ context) | 43.13% |
| State-tactic prompting | 31.37% |
| State-tactic tuning | 27.45% |
| File tuning | 41.18% |

Table 5: Performance comparison of different models on Math2001.

| Automation | File (%) | State-tactic (%) |
|---|---|---|
| Enabled | 41.18 | 11.76 |
| Disabled | 27.45 | 7.84 |

Table 6: Performance on the Math2001 split with and without access to standard automation.

to 2 theorems from each file (depending on the length of the file), for a total of 50 theorems. Of these, 31 have no proof in the Math2001 repository, hence testing theorem-level generalization.

**Context-aware models surpass state-based models**  Table 5 shows the performance comparison of different models. Both the GPT-4o model, which includes context in the input, and the file-tuned model perform significantly better than the other models. This demonstrates the importance of context information in context-dependent textbook-style problems.

**Models rely on common symbolic automation.**  The Math2001 split originally disables powerful automation tactics including `simp` and `nlinarith` to promote manual reasoning, akin to traditional textbook exercises. In Table 6 we evaluate models with the automation disabled, and observe substantial performance drops, confirming a heavy reliance of current models on these automation tactics. An examination of the training corpus further revealed a general dependency on automated tactics within real Lean projects, indicating that our models have learned to rely on these tactics.

# G  NTP-TOOLKIT and file-tuning details

## G.1  Data extraction

NTP-TOOLKIT contains a general-purpose data extraction tool that extracts examples from an arbitrary Lean 4 repository and formats them into examples that can be used to compile miniCTX, as well as for language-model fine-tuning. The tool is implemented in Lean based on Kim Morrison's `lean-training-data`.

Specifically, NTP-TOOLKIT takes in a configuration file with one or more Lean repositories specified. Each repository is transformed into *next-tactic* and *full proof* examples stored in JSON Lines files. The next-tactic data is suitable for making file-tuning examples of the form (context, state, next-tactic):

```
{
  "state": # tactic state ,
  "nextTactic": # pretty-printed next tactic,
  "srcUpToTactic": # source code in the file up to the tactic invocation,
  "decl": # declaration without proof (e.g., statement of a theorem),
  "declUpToTactic": # source code in the declaration up to the tactic invocation,
  "declId": # unique identifier of the declaration
}
```

The full proof data is suitable for making evaluation examples of the form (context, theorem, proof):

```
{
  "srcUpToDecl": # source code in the file up to the declaration,
  "decl": # declaration without proof (e.g., statement of a theorem),
  "declId": # unique identifier of the declaration,
  "proof": # proof
```

```
}
```

Full proof data is also suitable for training a model to directly generate a full proof, and NTP-TOOLKIT also provides Lean source with proof states interleaved, both of which we do not explore in this work.

## G.2   Input-output formatting.

Below we show the inputs and outputs for file-tuning and state-tactic tuning. In the paper we refer to the natural language description at the beginning of the input as an "instruction", and refer to a set of inputs and outputs as described below as "instruction-tuning data".

### G.2.1   File tuning.

Given an example containing a state, next-tactic, and preceding file contents (`srcUpToTactic`), the data is formatted as:

*Input*:

```
/- You are proving a theorem in Lean 4.
You are given the following information:
- The file contents up to the current tactic, inside [CTX]...[/CTX]
- The current proof state, inside [STATE]...[/STATE]

Your task is to generate the next tactic in the proof.
Put the next tactic inside [TAC]...[/TAC]
-/
[CTX]
{srcUpToTactic}
[/CTX]
[STATE]
{state}
[/STATE]
[TAC]
```

*Output*:

```
{nextTactic}
[/TAC]
```

### G.2.2   State-tactic tuning.

Given an example containing a state and next-tactic, the data is formatted as:

*Input*:

```
/- You are proving a theorem in Lean 4.
You are given the following information:
- The current proof state, inside [STATE]...[/STATE]

Your task is to generate the next tactic in the proof.
Put the next tactic inside [TAC]...[/TAC]
-/
[STATE]
{state}
[/STATE]
[TAC]
```

*Output*:

```
{nextTactic}
[/TAC]
```

### G.2.3   GPT-4o prompt

For full proof generation task with only theorem statement, we use the following prompt:

Your task is to generate complete proofs for problems stated in Lean4. You may use any tactics available in Mathlib, but no additional context, definitions, or theorems from the problem's file will be provided. Focus on crafting proofs using general knowledge and techniques applicable in Lean4. Here are some examples:

```
lemma deriv_scale {f : CS (n + 1) E} : (f.scale R).deriv = R⁻¹ ·
    f.deriv.scale R := by
  ext v ; by_cases hR : R = 0 <;> simp [hR, scale]
  · simp [deriv, smul] ; exact deriv_const _ _
  · exact ((f.hasDerivAt (R⁻¹ · v)).scomp v (by simpa using (hasDerivAt_id
    v).const_smul R⁻¹)).deriv

theorem mul_dvd_mul_left (a : α) (h : b | c) : a * b | a * c := by
  obtain ⟨d, rfl⟩ := h
  use d
  rw [mul_assoc]

/- Now here is your exercise. There is no need to restate the problem. If
    needed, think through the proof using comments. -/
{theorem statement}
```

For full proof generation task with additional infile context, we use the following prompt:

```
Your task is to generate complete proofs for problems stated in Lean4. For each problem, you
will be provided with the context from the file in which the theorem is stated. This context
includes useful external libraries, along with important definitions and theorems that are
relevant to the proof. You are encouraged to use any tactics, definitions, lemmas, or theorems
defined within this context to construct your proof. Please pay careful attention to indentation
and formatting to ensure that the proof adheres to Lean4 syntax standards. Here are some
examples:

#Context:
import Mathlib.Analysis.Calculus.Deriv.Support
import Mathlib.Analysis.Distribution.SchwartzSpace
import Mathlib.Order.Filter.ZeroAndBoundedAtFilter

open Real Complex MeasureTheory Filter Topology BoundedContinuousFunction
    SchwartzMap  BigOperators

variable {E : Type*} [NormedAddCommGroup E] [NormedSpace ℝ E] {{n : ℕ}}

@[ext] structure CS (n : ℕ) (E : Type*) [NormedAddCommGroup E] [NormedSpace
    ℝ E] where
  toFun : ℝ → E
  h1 : ContDiff ℝ n toFun
  h2 : HasCompactSupport toFun

noncomputable def scale (g : CS n E) (R : ℝ) : CS n E := by
  by_cases h : R = 0
  · exact ⟨0, contDiff_const, by simp [HasCompactSupport, tsupport]⟩
  · refine ⟨fun x => funscale g R x, ?_, ?_⟩
    · exact g.h1.comp (contDiff_const.smul contDiff_id)
    · exact g.h2.comp_smul (inv_ne_zero h)

/- Truncated -/

/- Now here is your exercise. There is no need to restate the problem. If
    needed, think through the proof using comments. -/
#Context:
{}

#Problem:
{}

{theorem statement}
```

## H  Additional results and analysis

### H.1  Dependency distribution

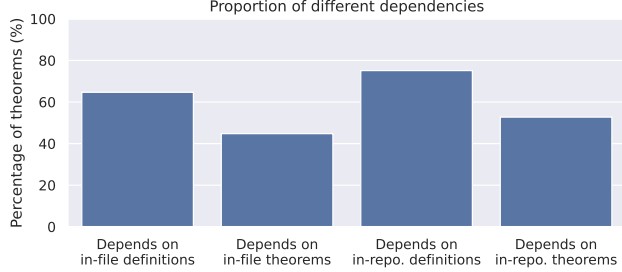

Figure 5: Percentage of different dependencies in the human-written proof of theorems in `miniCTX`

### H.2 Performance by proof length and context length

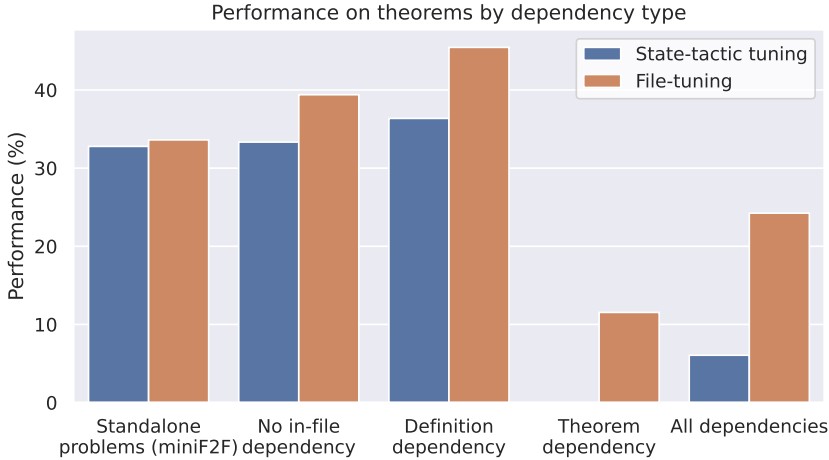

Figure 6: Performance by dependency type. For each theorem in `miniCTX`, we record as metadata whether its human-written proof depends on other definitions or theorems in the same file, and test the performance of baselines on each type. File-tuned models substantially outperform state-tactic tuned models on theorems with definition and/or theorem dependencies.

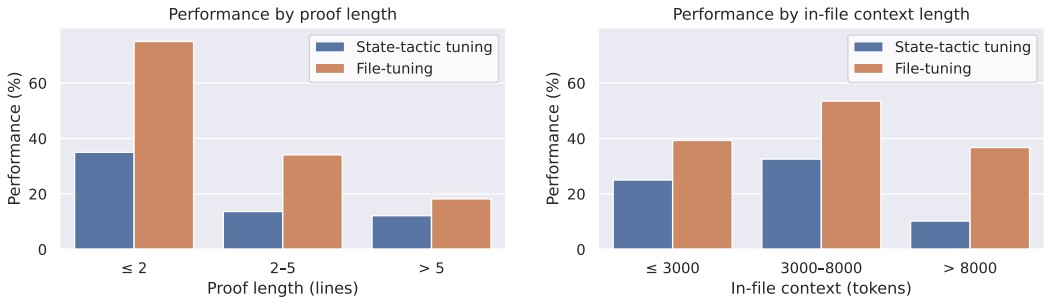

Figure 7: Performance of two baselines on different difficulty levels and context lengths, as measured by the length of human-written proof in lines and the size of the preceding file contents in tokens. File-tuning substantially improves theorem-proving abilities across all cases, but especially when the theorem is easier and the context is longer.

### H.3 Interference between in-file context and retrieved premises

In our experiments, we attempted to supply both in-file context (in the form of preceding code) and premise context (in the form of retrieved premises) to GPT-4o for proving a theorem. In Figure 8, we present an analysis of the impact of the length of retrieved premises on the resulting proof success rate.

**Longer retrieved premises hurt performance.** The results indicate that problems with a lower premise-to-context length ratio tend to have higher success rates. Specifically, successful problems often feature relatively shorter premises as proportion of the full context length. This suggests that models are better able to utilize and focus on relevant in-file context when the cross-file premises are proportionally smaller. Conversely, when the length of the premises becomes relatively large compared to the full context, it may overwhelm or distract the model, reducing its ability to effectively utilize the in-file information. This finding highlights the importance of ensuring a balanced integration of premises with the in-file context to maintain model focus and improve proof generation performance.

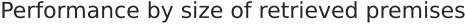

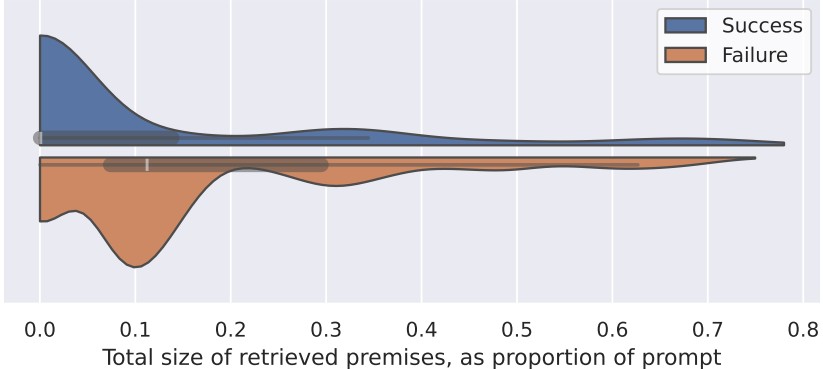

Figure 8: Impact of length of retrieved premises on GPT-4o model performance. A higher premise-to-context length ratio is correlated with lower success rates, suggesting that too much premise in context overwhelms the model.

## H.4  Example of learning proofs from context

The file-tuned model is able to utilize the proofs in the context. Here is an example of the model making minimal modification to the proofs from the context:

*Input*:

```
...
theorem Set.left_not_mem_uIoo {a b : ℝ} : a ∉ Set.uIoo a b :=
  fun ⟨h1, h2⟩ ↦ (left_lt_sup.mp h2) (le_of_not_le (inf_lt_left.mp h1))

theorem Set.right_not_mem_uIoo {a b : ℝ} : b ∉ Set.uIoo a b := by
```

*Output*:

```
    rintro ⟨h1, h2⟩
    exact (right_lt_sup.mp h2) (le_of_not_le (inf_lt_right.mp h1))
```

## H.5  Example of using theorems from context

The file-tuned model is able to utilize the stated theorems in the context. Here is an example of the model using the previously defined theorem in the proof:

*Input*:

```
...
lemma RectSubRect {x₀ x₁ x₂ x₃ y₀ y₁ y₂ y₃ : ℝ} (x₀_le_x₁ : x₀ ≤ x₁)
    (x₁_le_x₂ : x₁ ≤ x₂)
    (x₂_le_x₃ : x₂ ≤ x₃) (y₀_le_y₁ : y₀ ≤ y₁) (y₁_le_y₂ : y₁ ≤ y₂) (y₂_le_y₃
    : y₂ ≤ y₃) :
    Rectangle (x₁ + y₁ * I) (x₂ + y₂ * I) ⊆ Rectangle (x₀ + y₀ * I) (x₃ + y₃ *
    I) := by
  rw [rect_subset_iff, mem_Rect, mem_Rect]
  refine ⟨⟨?_, ?_, ?_, ?_⟩, ?_, ?_, ?_, ?_⟩
  all_goals simpa using by linarith

lemma RectSubRect' {z₀ z₁ z₂ z₃ : ℂ} (x₀_le_x₁ : z₀.re ≤ z₁.re) (x₁_le_x₂ :
    z₁.re ≤ z₂.re)
    (x₂_le_x₃ : z₂.re ≤ z₃.re) (y₀_le_y₁ : z₀.im ≤ z₁.im) (y₁_le_y₂ : z₁.im ≤
    z₂.im)
    (y₂_le_y₃ : z₂.im ≤ z₃.im) :
    Rectangle z₁ z₂ ⊆ Rectangle z₀ z₃ := by
```

*Output*:

```
simpa using RectSubRect x₀_le_x₁ x₁_le_x₂ x₂_le_x₃ y₀_le_y₁ y₁_le_y₂
    y₂_le_y₃
```

### H.6   Example of using unseen definitions

The state-tactic tuned model is able to utilize the unseen definitions that appear in the proof state. In the following example `Set.uIoo` is a newly defined definition, which is never seen for state-tactic tuned model:

*Input*:

```
theorem uIoo_of_le {α : Type*} [Lattice α] {a b : α} (h : a ≤ b) : Set.uIoo
    a b = Ioo a b := by

[STATE]
1 goal
zw: ℂ
c: ℝ
α: Type u_1
inst: Lattice α
ab: α
h: a ≤ b
⊢ uIoo a b = Ioo a b
[/STATE]
```

*Output*:

```
simp [uIoo, h]
```

This leads the state-tactic tuned model to perform similarly well as the file-tuned model on problems that only rely on new definitions, and not new theorems (see Figure 6).

# I Dataset hosting and maintenance

miniCTX is released on HuggingFace: l3lab/miniCTX, distributed under the Apache 2.0 license. Data extraction tool NTP-TOOLKIT is released on GitHub: cmu-l3/ntp-toolkit, under the MIT license. We note that the underlying data for the individual splits of miniCTX are also released under the Apache 2.0 license. We include the licensing information in the dataset repository. We plan to regularly update and maintain the dataset to include examples from new projects.

# J NTP-TOOLKIT guideline

We introduced NTP-TOOLKIT in §**??**. With the NTP-TOOLKIT, users can extract and annotate new theorems and proofs from any valid Lean project, in miniCTX format. The extracted data can be used either as updates to miniCTX, or as training data (for which we also provide instruction tuning utilities). We also develop a lightweight evaluation framework for easy evaluation on miniCTX.

## J.1 Preliminary

The evaluation code relies heavily on the Lean REPL [30], which operates within the project environment. Therefore, it is essential that the project builds without any errors. Additionally, the version of Lean used in the project should match the version supported by the REPL. While the Lean REPL supports versions $\geq 4.3.0$, for the best experience with data extraction and evaluation, we recommend evaluating projects that use Lean version 4.7.0 (all miniCTX theorems are in 4.7.0). We plan to continuously update NTP-TOOLKIT to support newer versions.

## J.2 Using the NTP-TOOLKIT

The NTP-TOOLKIT is designed to easily extract and annotate theorem proving data from Lean projects, by simply providing the project URL. To use the NTP-TOOLKIT for data extraction, follow these steps:

1. Installation: Clone the NTP-TOOLKIT repository from GitHub to your local machine, and checkout the Lean version tag corresponding to the extracted project (e.g., v4.7.0). Ensure that you have the required dependencies installed, as listed in the repository's README file.

2. Configuration: Supply GitHub URL, commit hash, and root modules of your Lean project in a JSON configuration file. Make sure that your project is using a compatible version of Lean. NTP-TOOLKIT will extract data from all modules imported by the root modules.

3. Data extraction: Run the data extraction script provided by the toolkit. Specify the `--full_proof_training_data` and `--premises` options to extract miniCTX-style data, which will be stored in an `minictx.jsonl` output file. Specify the `--declarations` option to additionally extract the premises in each module, for premise retrieval. The `full_proof_training_data` outputs can be additionally used for fine tuning (assuming the extracted data is dated before the current temporal split of miniCTX).

For detailed commands and additional options, please refer to the README file in the NTP-TOOLKIT repository.

## J.3 miniCTX Evaluation

We provide a comprehensive evaluation pipeline in the miniCTX-eval repository, supporting both tactic-prediction and full-proof generation tasks. Users should place the extracted JSONL file from the NTP-TOOLKIT into the data folder. To run an evaluation task, execute the task script by specifying the dataset path, the corresponding project path, and the path to the Lean REPL. This setup ensures that the evaluation is conducted within the correct environment and with the necessary data inputs.

