# OpenReview forum: "miniCTX: Neural Theorem Proving with (Long-)Contexts"
_NeurIPS.cc/2024/Workshop/MATH-AI — MATH-AI 24_

### Official Review · Reviewer_fQr9 · 2024-10-06
**Context-Dependent Theorem Proving with miniCTX: Valuable Contribution, but Needs Clearer Presentation**

**Rating:** 6
**Confidence:** 3

**Review:**

### **Summary and Final Decision**

The paper introduces the miniCTX dataset and NTP-TOOLKIT, making significant contributions to context-dependent theorem proving. The experiments demonstrate that providing models with context improves performance, particularly through file-tuning. However, explanations of key results, especially comparisons between miniCTX and miniF2F, need more depth. Additionally, the structure could be improved for better readability, and broader implications of the work could be explored more fully. Overall, the contributions are valuable but need clearer presentation of results and smoother transitions.

Rating: 6 (Marginally above acceptance threshold)

### **Strengths:**

1. **Innovative Benchmark (miniCTX):**
   The introduction of miniCTX as a dataset to test context-dependent theorem proving is a significant contribution. It addresses a critical gap in evaluating models' abilities to handle long, complex contexts, making it a valuable tool for future research in the field.

2. **Practical Toolkit (NTP-TOOLKIT):**
   The development of NTP-TOOLKIT for data extraction and evaluation is a strong point, enabling reproducibility and simplifying the process of extending the benchmark. This adds practical value and allows for further exploration by other researchers.

3. **Clear Experimental Setup:**
   The experimental setup is well-structured, comparing different approaches (file-tuning, state-tactic, GPT-4o) across datasets. The choice to compare models with and without context helps demonstrate the importance of leveraging background information in theorem proving.

---

### **Areas for Improvement:**

1. **Insufficient Explanation of Results:**
   The comparison between miniCTX and miniF2F, while valuable, lacks a clear explanation of why file-tuning performs significantly better in one case but not the other. More detailed analysis is needed to help readers understand the factors influencing these results.

2. **Limited Exploration of Broader Impacts:**
   The paper focuses heavily on theorem proving but misses an opportunity to discuss broader implications, such as how this approach could be applied to other domains like mathematical education or AI-driven mathematical discovery. Expanding the discussion would enhance the paper’s impact.

3. **Clarity of Language and Structure:**
   While the structure of the paper is generally sound, certain sections (particularly around the experimental results) could be more cohesive. Transitions between key concepts could be smoother, and the narrative could benefit from more detailed explanations, especially when discussing the significance of the results. Improving the flow would enhance readability and comprehension.

4. **Limited Quantitative Comparisons:**
   The paper mainly provides qualitative comparisons between datasets like miniCTX and miniF2F. Including more quantitative comparisons, such as how models perform on the same theorems across multiple datasets, would give a stronger empirical foundation to the claims.




### **Evaluation Criteria:**

1. **Relevance to the Journal (5/5):**
   The paper aligns very well with the journal's focus on AI and mathematical reasoning. It explores an important intersection—leveraging large language models (LLMs) for theorem proving—while addressing key challenges in context-dependent proving. This fits the theme of mathematical comprehension and reasoning capabilities of AI models.

2. **Clarity of Language and Structure (3/5):**
   The structure of the paper is generally clear, but some sections (e.g., explanations of results) would benefit from more detailed discussion. The transitions between key concepts, especially around the experimental results, could be improved for better readability and flow.

3. **Experimentation Procedure (3/5):**
   While the experiments are generally well-designed, some results (such as those on miniF2F) lack depth in explanation. It would be helpful if the authors provided more detailed insights into why file-tuning works well in some cases but not others. A more controlled comparison across datasets would strengthen the experimental procedure.

4. **Innovation and Original Contribution (4/5):**
   The introduction of the miniCTX benchmark is a valuable contribution, but the paper could go further in exploring real-world implications of context-dependent proving. The novelty of the toolkit is commendable, but expanding on potential applications and future use cases would increase the impact of the work.


### Line-By-Line Review

- **Line 3:** "We introduce miniCTX, which tests a model’s ability to prove formal mathematical theorems that depend on new context not encountered in training."

  **Comment:** Specify that miniCTX is a dataset to make this clearer. Without this, it may not be immediately obvious to the reader what miniCTX refers to.

---

- **Line 12:** "We found that this ability to use context is not captured by previous benchmarks such as miniF2F."

  **Comment:** Clarify whether miniF2F is a dataset. Consistently defining these terms is important for avoiding confusion later in the paper when multiple datasets are referenced.

---

- **Lines 61-62:** "Where (x, c) ∼ p denotes a (theorem, context) pair from a context distribution p, M is a model that produces a proof y, and v returns 1 if the proof is correct and 0 otherwise."

  **Comment:** The text only mentions evaluating fully correct proofs, but what about partial proofs? Including an evaluation of partial proofs would provide more insights into model performance, especially for complex problems.

---

- **Line 70:** "Table 1 provides a qualitative comparison with other datasets."

  **Comment:** The table provides a good qualitative comparison, but it would be interesting to see a **quantitative comparison** as well. For example, if the same theorem appears across multiple datasets, it would be valuable to compare how well models trained on miniCTX perform relative to other datasets. This would strengthen the empirical evidence in the paper.

---

- **Lines 124-136:** "We see a dramatic improvement for the file-tuned model over the state-tactic methods (31.65% vs. 9.31%). Similarly, providing preceding file context to GPT-4o results in dramatic improvement (22.07% vs. 5.59%)."

  **Comment:** The results don't provide enough insight into why file-tuning performs much better on miniCTX but not on miniF2F. This comparison between datasets feels incomplete. A better comparison would be to evaluate multiple datasets using the same test data and show how miniCTX excels at proving theorems under those conditions.

---

- **Line 131:** "The file-tuned model improves very little beyond the state-tactic model (33.61% vs. 32.79%) on miniF2F."

  **Comment:** It’s unclear what this small improvement on miniF2F is meant to prove. The authors should explain why file-tuning does not yield similar gains in miniF2F as it does in miniCTX, or clarify the significance of this result.

---

### Official Review · Reviewer_h1o2 · 2024-10-06
**Positive contribution to the community, clear acceptance**

**Rating:** 8
**Confidence:** 5

**Review:**

## Strengths

- The work provides a robust evaluation dataset for neural theorem proving, which is very much needed in the community, as previous evaluation suites had serious issues (not only regarding context, but also contamination).
  - The authors utilize knowledge cutoffs to robustly evaluate models' performance without worrying about contamination, and include preceding file contents to better specify the intent.
- Both the eval and train sets are quite large (583k train, 15k dev, and 15k test), allowing for extensive evaluation and training.
- The authors release their code and dataset, making it a valuable contribution that can be used by the community.

## Weaknesses

- Figures could have been made crisper. Figure 1 is hard to read without zooming in, and the lack of vectorization makes it appear pixelated.
- The paper lacks hyperpameters used for both inference and training, making it hard to reproduce results.

---

### Official Review · Reviewer_6ysy · 2024-10-07
**Solid work on new benchmarks but could use some improvement**

**Rating:** 6
**Confidence:** 3

**Review:**

### Summary:

This work introduces a benchmark to evaluate automated theorem provers when a local context (e.g., repository-wide) is necessary, as well as a supporting tool to extract and update the benchmark to ensure it stays uncontaminated.

### Strengths:

This work focuses on a benchmark that requires a strong understanding of the context.

This work can be updated continuously to prevent data contamination.

### Limitations:

There are few model evaluations (GPT-4o and Deepseek Coder 1.3B) only.

The benefits over other benchmarks are not extremely clear.

---

### Official Review · Reviewer_AVmJ · 2024-10-07

**Rating:** 7
**Confidence:** 4

**Review:**

Summary: In this submission, the authors recognize as a problem that many neural theorem-proving works are geared to competition-problem solving, but this does not reflect the kind of formalization work actually done by mathematicians. Further, this formalization work is heavily dependent on previously written lemmas and definitions. Other works usually do not provide these as direct context to the model, instead expecting an understanding to be learned during training. However, directly working with background context better reflects a mathematician's formalization workflow. They produce a new benchmark based on testing models ability to operate with context on a number of prior formalization projects, including research-level repositories like PFR and PNT. They also perform some experiments on the benchmark like LLM prompting, BFS with state-tactic prediction, and file-tuning.

Significance: Many formalization projects span many files with many definitions that have been carefully constructed for the particular task. To automate the formalizing workflow, a model must be aware of all these definitions but this can be difficult with a model that has not been trained on all this context.

1. The authors identify a valuable problem and produce a benchmark that is relevant for the overall task. It is also time-robust and can be updated mostly automatically (clarification: when being updated you will still need to choose new repositories to include in the benchmark, right?) The benchmark also has a varied difficulty, which can be useful for testing out different kinds of models.
2. The experiments are useful for building an intuition for how popular approaches perform with and without context. It should be relevant to further explorations of the problem statement.

Overall, I judge this work to be valuable, especially for the AI-4-math community.

---

### Decision · Program_Chairs · 2024-10-07

Accept